# Storm surge dynamics in the northern Adriatic Sea: comparing AI emulators with high-resolution numerical simulations

Rodrigo Campos-Caba<sup>1,2</sup>, Paula Camus<sup>3</sup>, Andrea Mazzino<sup>4,5</sup>, Michalis Vousdoukas<sup>6,7</sup>, Massimo Tondello<sup>8</sup>, Ivan Federico<sup>9</sup>, Salvatore Causio<sup>9</sup>, and Lorenzo Mentaschi<sup>1,2,9</sup>

- 1. Department of Physics and Astronomy (DIFA), University of Bologna, 40127 Bologna, Italy
- 2. Interdepartmental Research Centre for Environmental Sciences (CIRSA), University of Bologna, 48123 Ravenna, Italy
- 3. Departamento de Ciencias y Técnicas del Agua y del Medio Ambiente, University of Cantabria, 39005 Santander, Spain
- 4. Department of Civil, Chemical and Environmental Engineering, University of Genoa, 16145 Genoa, Italy
- 5. Istituto Nazionale di Fisica Nucleare, Sezione di Genova, 16146 Genoa, Italy
  - 6. Department of Marine Sciences, University of the Aegean, 81100 Mitilene, Greece
  - 7. MV Coastal and Climate Research Ltd., 3046 Limassol, Cyprus
  - 8. HS Marine SrL, 35027 Noventa Padovana, Italy
  - 9. CMCC Foundation Euro-Mediterranean Center on Climate Change, 73100 Lecce, Italy
- Correspondence to: Rodrigo Campos-Caba (rodrigo.camposcaba@unibo.it), Lorenzo Mentaschi (lorenzo.mentaschi@unibo.it)

Abstract. Accurate storm surge forecasting is vital for protecting coastal regions, particularly in the northern Adriatic Sea where sea-level rise and increasingly severe storm events pose growing risks. Machine Learning (ML) approaches offer compelling speed and flexibility, yet their ability to emulate high-resolution dynamic models, especially for extreme surge events, has not been sufficiently assessed across methods and loss functions. In this study, a range of ML emulators, from Multivariate Linear Regression (MLR) to Long Short-Term Memory (LSTM) networks, is benchmarked against a high-resolution hydrodynamic model optimized for extreme surge representation. We also evaluate the impact of training loss functions, comparing the conventional Mean Squared Error (MSE) with the corrected Mean Absolute Deviation squared (MADc²), designed to better capture surge peaks. Results show that even simple models like MLR, when trained with MADc², achieve performance comparable to advanced neural networks while remaining orders of magnitude faster. These findings demonstrate that with appropriate training strategies, data-driven emulators can rival physics-based models in reproducing extremes. The MLR-MADc² configuration emerges as a practical balance between computational efficiency and accuracy, underscoring the potential of ML emulators for coastal forecasting and risk assessment.

© Author(s) 2025. CC BY 4.0 License.

## 1 Introduction

Accurate storm surge prediction is vital for coastal risk management, particularly in the context of climate change, which is intensifying the frequency and severity of extreme events (Calafat et al., 2022). While physics-based hydrodynamic models remain the standard for operational forecasting, recent developments in Machine Learning (ML) offer alternatives that are both computationally efficient and adaptable (Chen et al., 2022; Zhao et al., 2024). In this context, ML models trained to approximate the behavior of complex dynamic models or observed phenomena are conventionally called emulators, i.e. surrogate systems that replicate the input-output relationships without resolving the underlying physical processes.

Regression-based ML emulators, such as Multivariate Linear Regression (MLR), have been widely used to predict storm surge levels (Mi Zhang et al., 2019). More flexible architectures like Neural Networks (NNs), particularly Recurrent Neural Networks (RNN) and gated variants such as Long Short-Term Memory (LSTM) models, are especially well suited for time series forecasting, as they can capture both short- and long-term dependencies in surge dynamics (Bezuglov et al., 2016; Suradhaniwar et al., 2021). These emulators have demonstrated strong performance across a variety of coastal settings (Igarashi & Tajima, 2021; Chen et al., 2022; Wang et al., 2021; Adeli et al., 2023).

Tiggeloven et al. (2021) tested CNN-LSTM hybrid emulators but ultimately found that standard LSTM models outperformed more complex architectures. Further enhancements include Gated Recurrent Units (GRUs) with physics-informed loss functions (Feng & Xu, 2024) and transformer-based models (Rus et al., 2023b), which extend emulator capabilities in capturing both spatial and temporal surge patterns. Ensemble techniques and bias correction approaches have also contributed to improved predictive skill and interpretability (Giaremis et al., 2024; Sun & Pan, 2023).

In the Adriatic Sea, applications of ML emulators remain limited. Notable efforts include the integration of NNs into operational surge systems for Venice (Bajo & Umgiesser, 2010), and the HIDRA model series for Koper, Slovenia, which have outperformed coarser hydrodynamic models in short-term forecasts and in reconstructing surge events (Žust et al., 2021; Rus et al., 2023a,b).

Nevertheless, gaps still persist in the literature: (1) a general lack of systematic comparisons between ML emulators and high-resolution storm surge models developed specifically for the extremes; (2) limited consideration of how emulator performance may vary across different coastal environments; and (3) insufficient analysis of emulator skill during extreme events. These limitations highlight the need for careful calibration and validation of ML-based emulators in each coastal setting, in order to identify the most suitable approach for local conditions and to assess when such emulators can serve as viable alternatives to dedicated dynamic models, particularly for extremes.

© Author(s) 2025. CC BY 4.0 License.

EGUsphere Preprint repository

To help address these issues, this study benchmarks a range of ML-based emulators, from MLR to LSTM architectures, against a dedicated high-resolution hydrodynamic model of the northern Adriatic Sea. Emphasis is placed on assessing under which conditions emulators can reproduce the statistical and physical characteristics of storm surge, with a focus on their ability to capture extremes.

#### 2 Data and methods

In this study, we compare the performance of a high-resolution dynamic downscaling model with that of machine learning emulators of varying complexity. The remainder of this section outlines the numerical model and its forcing, the AI methods employed, and the datasets used for training and evaluation.

# 2.1 High-resolution numerical simulation

The numerical simulation used for comparison with the ML emulators is a dynamic downscaling of storm surges in the northern Adriatic Sea, based on the SHYFEM-MPI hydrodynamic model. SHYFEM-MPI is an unstructured-grid finite element code that solves the Navier-Stokes equations under hydrostatic and Boussinesq approximations (Umgiesser et al., 2004; Micaletto et al., 2022). It is an established modeling framework, previously applied in operational (Federico et al., 2017), relocatable (Trotta et al., 2016), and storm surge forecasting systems (Park et al., 2022; Alessandri et al., 2023).

The simulation employs an unstructured grid with a horizontal resolution of approximately 3 km at the open boundary and 50 m nearshore. Atmospheric forcing is provided by a 3.3 km downscaling of the Climate Forecast System Reanalysis (CFSR), performed using the WRF model. The simulation, hereinafter referred to as SHYFEM-MPI, spans the period 1987–2020 and provides hourly output. For more information, the reader is referred to Campos-Caba et al., (2024).

## 2.2 Predictand

Observed data from tide gauges were used as a predictand. The ML emulators were applied at Punta della Salute and Trieste (Figure 1), locations that offer consistent and long-term hourly observed data, a crucial requirement for training the implemented NN models. In both locations the data used extends from 1987 to 2020. Observations for Punta della Salute were provided by the Italian National Institute for Environmental Protection and Research (ISPRA), while data for Trieste were obtained from Raicich (2023). Additionally, data from the ISMAR-CNR research platform "Aqua Alta" (hereafter CNR platform, Figure 1) were considered to compare the ML models with the regional numerical model for a storm surge event that occurred in November 2022 (Mel et al., 2023).

Figure 1: Study area and locations for the implementation of the machine learning models. (a) Overview of the Adriatic Sea. (b) Northern Adriatic study area showing the locations where the ML emulators were applied, along with the ISMAR-CNR platform.

## 2.3 Predictors

The predictors used for ML emulators include meteorological and oceanographic variables known to influence storm surge dynamics and commonly adopted in recent data-driven studies (e.g., Kim et al., 2019; Žust et al., 2021; Chen et al., 2022; Rus et al., 2023a; Tausía et al., 2023; Harter et al., 2024; Dang et al., 2024). Specifically, we used sea surface height from Med-MFC (Escudier et al., 2021), tides from FES2014 (Lyard et al., 2021), and mean sea level pressure from ERA5 (Hersbach et al., 2020). Following Harter et al. (2024), wind stress was also included, computed from ERA5 wind fields. ERA5 variables were interpolated to the Med-MFC grid using the method by Wang et al. (2024).

The spatial domain for predictor extraction was selected via performance testing with multivariate linear regression (MLR) emulators and is shown in the Supplementary Information (Figure S1). Since most predictors are spatial fields and the target is time-series regression, dimensionality reduction was applied using Principal Component Analysis (PCA). The first seven principal components (PCs) of each predictor were used independently, without merging into a multivariate series. This number was selected based on MLR performance tests. Tidal data were excluded from this process and used directly as single

hourly time series. The selected EOFs for each predictor are shown in Figures S2–S5 and the explained variance of each PC for is shown in Figure S6.

To assess the relative influence of the input features on emulator predictions, a permutation importance analysis was conducted. This approach measures the contribution of each feature by randomly shuffling its values across the test dataset, thereby breaking its relationship with the target, while leaving all other features unchanged. The trained emulator then generates predictions on the permuted dataset, and the resulting mean absolute deviation (MAD) is compared with the baseline MAD obtained from the original inputs. The increase in MAD reflects the feature's importance, with larger increases indicating a stronger influence on predictions. To mitigate randomness, each permutation was repeated 10 times, and the final importance score was computed as the average increase in MAD. This procedure offers a model-agnostic and interpretable assessment of feature relevance, enabling the identification of the most influential predictors driving the emulator's performance.

## **120 2.4 Emulators**

The ML emulators were implemented in Python using the PyTorch library (Paszke et al., 2017), a widely adopted framework for AI applications. The emulators used in this study range from simple approaches such as Multivariate Linear Regression (MLR) to deep recurrent architectures, including Long Short-Term Memory (LSTM) networks (Table 1).

Table 1: List of emulator architectures employed in this study.

| ML model                                  | Description                                                                                                                                                                                                                                                                                                                                                                                                                                          |  |  |
|-------------------------------------------|------------------------------------------------------------------------------------------------------------------------------------------------------------------------------------------------------------------------------------------------------------------------------------------------------------------------------------------------------------------------------------------------------------------------------------------------------|--|--|
| Multivariate Linear Regression (MLR)      | The MLR emulator captures the linear relationship between predictors and the predictand by estimating coefficients via least squares minimization.                                                                                                                                                                                                                                                                                                   |  |  |
| Multilayer Perceptron (MLP)               | The MLP is a fully connected feedforward neural network with two hidden layers of 120 neurons each, using ReLU activations to capture non-linear patterns in time-series regression. The output layer generates the final predictions.                                                                                                                                                                                                               |  |  |
| Simple Recurrent Neural Network (RNN)     | The simple Recurrent Neural Network (RNN) is designed to process sequential data by maintaining a hidden state that evolves over time. In this study, the RNN consists of a single hidden layer with 120 units. At each time step, the hidden state is updated based on the current input and the previous hidden state, enabling the model to learn temporal dependencies. The output layer maps the final hidden states to the target predictions. |  |  |
| Hybrid Recurrent Neural Network<br>(RNNh) | The RNNh emulator combines a traditional RNN layer with a parallel linear layer to enhance feature learning. The RNN layer includes 60 hidden units, while the linear layer independently transforms the input features using ReLU activation. The outputs of both layers are concatenated and passed to the prediction stage, allowing the model to capture both temporal dependencies and enriched input representations.                          |  |  |

| Long Short-Term Memory<br>(LSTM)         | The LSTM emulator includes a Long Short-Term Memory layer with 120 hidden units. LSTM use internal gating mechanisms (input, forget, and output gates) to regulate the flow of information over time. These gates enable the model to retain relevant patterns across long sequences or discard outdated inputs, making it well suited for time series with variable temporal dependencies. |  |
|------------------------------------------|---------------------------------------------------------------------------------------------------------------------------------------------------------------------------------------------------------------------------------------------------------------------------------------------------------------------------------------------------------------------------------------------|--|
| Hybrid Long Short-Term Memory<br>(LSTMh) | The LSTMh emulator combines an LSTM layer (60 units) with a parallel linear layer to en predictive accuracy. The LSTM processes the sequential input, while the linear independently transforms the features using ReLU activation. The two outputs are concatenated, allowing the model to leverage both long-term dependencies and enriched frepresentations.                             |  |

The depth and size of the hidden units for the NN emulators (MLP, RNN, and LSTM) were selected through an exploratory hyperparameter search, where multiple configurations were tested and the one yielding the best validation performance, as measured by the MSE, was adopted as the optimal setup for each algorithm. As loss functions both MSE and MADc<sup>2</sup>, were applied across all models. The MADc<sup>2</sup> loss is a recently introduced metric specifically designed to improve the representation of extremes; its full definition and properties are detailed in Section 2.7.

The adoption of the MADc<sup>2</sup> loss function warrants a brief discussion from both theoretical and practical perspectives. In this section, we focus on its definition as the square of MADc and its convexity, that is, the existence of a single global minimum, which underpins the robustness of gradient-based optimization algorithms. MADc<sup>2</sup> is preferred over MADc because it is differentiable at the theoretical minimum (MADc<sup>2</sup> = 0), whereas MADc is not. This differentiability enables smoother convergence in optimization routines, particularly those that rely on gradient information.

Theoretically, MADc² is convex: for an ideal model that perfectly reproduces the observations, both the median absolute deviation (MAD) and the prediction bias term (MADp) vanish. The loss function reaches its unique global minimum at this point of perfect agreement, with no other minima admissible. In practice, however, convexity may be locally attenuated if the parameter values that minimize MAD differ from those minimizing MADp. In such cases, compensation effects between the two components can generate shallow valleys or low-gradient regions in parameter space, which may slow down convergence.

To empirically assess the suitability of MADc<sup>2</sup> as a loss function, we analyzed its behavior in a synthetic case study involving a simple univariate linear regression with one weight (W) and one bias (B). The resulting loss surface displays a well-defined and narrow minimum region near the theoretical optimum values W = 2 and B = 1 (Figure S7), supporting the practical usability of MADc<sup>2</sup> in typical regression scenarios.

Considering the different configurations and loss functions, 12 ML emulators were implemented (Table 2).

Table 2: List of the different ML emulators implemented.

| ML emulator configuration | Loss function     | Acronym                  |
|---------------------------|-------------------|--------------------------|
| MLR model                 | MSE               | MLR-MSE                  |
| MLP model                 |                   | MLP-MSE                  |
| RNN model                 |                   | RNN-MSE                  |
| RNNh model                |                   | RNNh-MSE                 |
| LSTM model                |                   | LSTM-MSE                 |
| LSTMh model               |                   | LSTMh-MSE                |
| MLR model                 | $\mathrm{MADc^2}$ | MLR- MADc <sup>2</sup>   |
| MLP model                 |                   | MLP- MADc <sup>2</sup>   |
| RNN model                 |                   | RNN- MADc <sup>2</sup>   |
| RNNh model                |                   | RNNh- MADc <sup>2</sup>  |
| LSTM model                |                   | LSTM- MADc <sup>2</sup>  |
| LSTMh model               |                   | LSTMh- MADc <sup>2</sup> |

## 155 2.6 Training, validation, and testing

To prepare the data for training and testing the ML emulators, both the predictand and predictors were standardized by subtracting the mean and dividing by the standard deviation, resulting in zero mean and unit variance. This normalization ensures that each predictor contributes equally to the learning process, preventing variables with larger magnitudes from disproportionately influencing model training.

Following the approach of Gholamy et al. (2018), 80% of the available data (28 years) was allocated for training, while the remaining 20% (6 years) was evenly split into validation and testing sets (3 years each). The specific years assigned to each set were as follows: training: 1987–1992 and 1997–2018; validation: 1993, 1994, and 2019; testing: 1995, 1996, and 2020. This temporal partitioning was designed to ensure that the distributions of observed storm surge values were comparable across sets, particularly between training and testing, in line with the recommendations of Uçar et al. (2020). Figure S8 illustrates this: panels (a) and (c) show that the training sets for Punta della Salute and Trieste include the highest observed percentiles, allowing the models to learn from extreme events. Panels (b) and (d) demonstrate that, for both locations, the testing sets also

© Author(s) 2025. CC BY 4.0 License.

include high-end extremes, facilitating a robust evaluation of the models' ability to reproduce rare but significant storm surge events.

Gradient-based training methods introduce randomness into the optimization process, so the final parameter set is not identical across different training runs. For this reason, each ML emulator was trained 40 times, with each run corresponding to a unique random initialization of model weights. For the NN emulators, each run consisted of 800 training epochs, while the MLR emulator was trained for 10000 iterations to ensure convergence. Model parameters were optimized using the Adam algorithm. The multiple runs allow assessment of variability due to weight initialization and provide the basis for selecting the best-

performing model.

Model selection was based on validation performance, using a validation-based model selection strategy, a widely adopted approach for identifying optimal configurations during training (Bishop, 2006; Goodfellow et al., 2016). Several performance metrics were tested on the validation set, including both general metrics over the entire period and specialized ones computed over storm surge events exceeding the 99th percentile. Among these, the slope of the linear fit between emulator output and observed values emerged as the most effective criterion, as it quantifies the model's ability to reproduce the dynamic range and intensity of storm surges, especially avoiding systematic under- or overestimation of extreme values. This slope was therefore used to identify the best-performing model run, which was then evaluated on the independent testing set to assess generalization performance.

# 2.7 Model performance evaluation

The ML emulators were evaluated against SHYFEM-MPI over the testing period. Additionally, the trained ML emulators were applied and compared with the Copernicus Sea Physics Analysis and Forecast product (hereinafter Med-Physics, Clementi et al. 2023) for a storm surge event that occurred in November 2022 in the northern Adriatic Sea (Mel et al., 2023). The performance assessment was based on statistical metrics computed from hourly data, considering the entire dataset and surge peaks above the 99th percentile of the predictand's cumulative distribution at each location. The statistical metrics used for evaluation are the following:

Slope of linear fit (SLF):

$$195 \quad S = m \ O + b \tag{1}$$

The slope of the linear fit m quantifies the scaling agreement between simulated (S) and observed (O) values. It is obtained by fitting a simple linear regression line to the scatterplot of emulators' output (simulated data) versus observations. A slope close to 1 indicates that the model accurately reproduces the magnitude and dynamic range of the target variable, while deviations from 1 reflect systematic over- or underestimation of variability.

Pearson correlation (Corr):

$$\rho = \frac{1}{N-1} \sum_{i=1}^{N} \left( \frac{S_i - \mu_S}{\sigma_S} \right) \left( \frac{O_i - \mu_O}{\sigma_O} \right) \tag{2}$$

The Pearson correlation coefficient ( $\rho$ ) quantifies the degree of linear dependence between simulated and observed data, with values closer to 1 indicating stronger agreement and better model performance. In the formula,  $\mu_S$  and  $\mu_O$  are the simulated and observed means, while  $\sigma_S$  and  $\sigma_O$  correspond to their standard deviations.

Root-Mean Squared Error (RMSE):

$$RMSE = \sqrt{\frac{1}{N} \sum_{i=1}^{N} (S_i - O_i)^2}$$
 (3)

The Root Mean Squared Error (RMSE) measures the average magnitude of the errors between simulated and observed values.

It is defined as the square root of the mean of the squared differences between simulations and observations. A value closer to zero indicates a better performance.

Bias:

225

230

205

$$Bias = \bar{S} - \bar{O} \tag{4}$$

Bias quantifies the systematic error (the mean difference) between simulated and observed values. Scores near 0 reflect minimal bias; positive values indicate overestimation, negative values underestimation. In the formula,  $\bar{S}$  and  $\bar{O}$  are the mean simulated and observed series. Since both datasets were detrended and mean-centered, bias was computed only for storm-surge peaks exceeding the 99th percentile, where residual offsets matter most.

220 Mean Absolute Deviation (MAD):

$$MAD = \overline{|S - 0|} \tag{5}$$

The Mean Absolute Deviation (MAD) measures the average absolute difference between simulated and observed values. It reflects overall prediction accuracy and is less sensitive to outliers than RMSE. A value closer to 0 indicates a better performance.

MAD of the percentiles (MADp) (Campos-Caba et al., 2024):

$$MADp = \overline{|S_{prc} - O_{prc}|} \tag{6}$$

MADp measures the average absolute error between simulated and observed values across percentiles of the observed distribution. It quantifies how well a model reproduces the empirical distribution of the target variable. A value closer to 0 indicates a better performance.

Corrected MAD (MADc) (Campos-Caba et al., 2024):

240

250

260

$$MADc = \overline{|S - O|} + MADp \tag{7}$$

MADc is a composite error metric combining the mean absolute deviation (MAD) with its percentile-based version (MADp).

It captures both overall magnitude errors and differences in the distribution of values, making it robust to phase shifts and effective for evaluating extremes. A value closer to 0 indicates a better performance.

As noted earlier, each emulator architecture was trained and evaluated on the testing set 40 times. For each run and location, statistical metrics were computed separately and then averaged across Punta della Salute and Trieste, yielding one set of values per run. Analyses were conducted on both the full dataset and on surge peaks above the 99th percentile of the predictand distribution. The full dataset was assessed using RMSE, MADp, and MADc, while surge peaks were evaluated with bias, MADp, and MADc. Section 3 displays violin plots of the resulting distributions across the 40 runs for each emulator—loss function combination.

To further evaluate emulator performance relative to the SHYFEM-MPI baseline, we employed a paired bootstrap procedure to quantify the statistical significance of differences in key error metrics (bias, MADp, and MADc). The bootstrap analysis was applied to surge peaks above the 99th percentile, using 10,000 resampling iterations and a significance level of 0.05. This approach provides 95% confidence intervals for the differences in metrics, allowing assessment of whether emulator errors systematically diverge from those of SHYFEM-MPI.

Complementing the bootstrap analysis, a one-sample t-test was performed on the vector of metric differences between the emulator and SHYFEM-MPI across all runs. The null hypothesis assumes that the mean difference is zero, corresponding to no systematic performance gain or loss. The test was applied at a significance level of 0.05, with negative mean differences indicating improved emulator performance and positive mean differences indicating deterioration relative to SHYFEM-MPI.

## **255 3 Results**

## 3.1 Performance evaluation on the full time series

The RMSE distributions across the 40 training experiments indicate that the ML-MSE emulators consistently outperform both the SHYFEM-MPI benchmark and most ML-MADc<sup>2</sup> emulators in terms of RMSE in the testing set. Among them, the simple MLR-MSE emulator ranks among the top performers in terms of RMSE (Figure 2a). While most MADc<sup>2</sup>-based emulators also surpass SHYFEM-MPI on this metric, the LSTM-MADc<sup>2</sup> variant achieves the best performance within its group. Regarding MADp, the MADc<sup>2</sup>-trained emulators exhibit performance comparable to SHYFEM-MPI, with a significant portion of their distributions outperforming the benchmark (Figure 2b). In contrast, the MSE-based emulators underperform in terms of MADp, particularly the MLR-MSE model, which exhibits the weakest alignment in the percentile–percentile comparison. Notably, the emulator that achieves the best RMSE score is also the one that reproduces the empirical distribution of the

reference sample most poorly, underscoring the trade-off between global error minimization and accurate representation of extremes (Figure 2b). For MADc, emulators trained with the MADc² loss consistently achieve lower errors than their MSE-trained counterparts, although their performance shows greater variability. This variability is especially pronounced among the RNN-based architectures, likely reflecting their known susceptibility to vanishing gradient issues (Figure 2c).

Figure 2: Violin plots of RMSE (top panel), MADp (middle panel), and MADc (bottom panel), averaged across locations for the 40 training runs. Each violin shows the distribution of metric values across runs for a given emulator. The white dot indicates the median, the thick black bar shows the interquartile range, and red stars denote the scores of the selected run (chosen based on the slope criterion). The black dashed line corresponds to the performance of SHYFEM-MPI.

© Author(s) 2025. CC BY 4.0 License.

# 275 3.2 Performance evaluation on the peaks above the 99th percentile

Focusing on surge peaks above the 99th percentile, the skill distributions across 40 training experiments in the testing set show that emulators trained with the MADc<sup>2</sup> loss function consistently outperform both the MSE-trained models and the SHYFEM-MPI benchmark in terms of bias, MADp, and MADc (Figure 3). The MLR-MADc<sup>2</sup> and LSTMh-MADc<sup>2</sup> models emerge as the top performers across all three metrics. In contrast, MLR-MSE, despite its strong RMSE performance on the full time series, performs the worst on the extremes. These differences highlight that MADc<sup>2</sup>-trained models are better suited to capturing the distributional characteristics of extreme surge events, where accuracy matters most.

Considering performance metrics averaged over the best-performing runs (those with SLF values closest to 1) the MADc<sup>2</sup> loss function consistently enhances MLR emulator performance in capturing extreme events. This improvement is evident across all evaluated metrics: RMSE, bias, MAD, MADp, and MADc. The MLR emulator shows the most substantial gains, with improvements of 24.09% in SLF, 35.79% in RMSE, 60.32% in bias, 40.00% in MAD, 56.56% in MADp, and 47.99% in MADc (Figure 4), elevating its performance to levels comparable to more complex architectures. Significant improvements are also observed in the LSTMh model (15.56% in RMSE, 44.75% in bias, 16.83% in MAD, 43.50% in MADp, and 31.17% in MADc), as well as in the RNNh (22.73% in RMSE, 33.16% in bias, 25.58% in MAD, 32.24% in MADp, and 28.46% in MADc) and the MLP model (13.64% in RMSE, 28.49% in bias, 13.12% in MAD, 27.32% in MADp, and 19.80% in MADc) (Figure 4).

295

280

285

Figure 3: Violin plots of RMSE (top panel), MADp (middle panel), and MADc (bottom panel), averaged across locations for the 40 training runs, for the surge peaks above the 99<sup>th</sup> percentile. Each violin shows the distribution of metric values across runs for a given emulator. The white dot indicates the median, the thick black bar shows the interquartile range, and red stars denote the scores of the selected run (chosen based on the slope criterion). The black dashed line corresponds to the performance of SHYFEM-MPI.

Figure 4: Mean percentage variation in performance metrics for surge peaks above the 99<sup>th</sup> percentile for the selected runs when the MADc<sup>2</sup> loss function is used to train the emulator. Positive values (blue colors) indicate an improvement in performance, while negative values (warm colors) indicate a decline.

T-test comparison of emulator vs SHYFEM-MPI performance (Figure 5) further confirms the advantage of the MADc<sup>2</sup>-trained emulators for extreme surge peaks representation. For the MLR and LSTMh emulators, statistically significant improvements are observed across bias, MADp, and MADc when trained with MADc<sup>2</sup>. In particular, both MLR-MADc<sup>2</sup> and LSTMh-MADc<sup>2</sup> exhibit mean differences that are consistently negative (favoring the emulator over SHYFEM-MPI), with confidence intervals excluding zero, highlighting their robustness in reducing systematic bias and error in extreme surge representation. By contrast, the MSE-trained models perform less consistently. MLR-MSE shows clear positive differences, indicating worse performance relative to SHYFEM-MPI in all three metrics, despite its strong skill on the full time series. LSTMh-MSE performs moderately better but still fails to achieve the same level of statistical robustness as the MADc<sup>2</sup>-trained counterpart.

Figure 5: Paired t-test results of emulator performance relative to SHYFEM-MPI for surge peaks above 99th percentile, averaged across locations. Metrics shown are bias, MADp, and MADc. Circles indicate mean differences; error bars show 95% confidence intervals. Green markers denote statistically significant improvements (p < 0.05), while gray markers indicate no significance. (a) and (c) MSE-trained emulators; (b) and (d) MADc<sup>2</sup>-trained emulators.

## 3.3 Performance evaluation at selected peaks

330

335

325 To further assess emulator performance under real-world conditions, we examined two high-impact periods from the testing set: November 1996 and December 2020, both marked by intense surge activity (Figure 6). These case studies reveal how the selected emulators (MLR-MADc² and LSTMh-MADc²) respond to complex surge dynamics, highlighting not only their capabilities but also specific limitations—for example, a tendency to reproduce dominant peaks more reliably than secondary structures, and site-dependent variability in amplitude reconstruction.

During the November 18, 1996, event the MLR-MADc<sup>2</sup> model accurately captures the surge amplitude at both locations, demonstrating that even a simple linear architecture can perform effectively on extremes when paired with a suitable loss function. The LSTMh-MADc<sup>2</sup> model, while performing well in Trieste, significantly underestimates the peak at Punta della Salute, suggesting a potential mismatch between the temporal structure of the model and the specific surge dynamics at that site. Nevertheless, both emulators align more closely with observations than the SHYFEM-MPI simulation, not only at the peak but also throughout the buildup and decay phases, indicating improved phase coherence and temporal fidelity.

Figure 6: Storm surge time-series for November 1996 and December 2020 in Punta della Salute and Trieste for observations, SHYFEM-MPI, and ML models MLR-MADc<sup>2</sup> and LSTMh-MADc<sup>2</sup>.

The December 2020 sequence, which includes two surge peaks (December 8 and 28), presents a more complex test. For the first peak, SHYFEM-MPI overestimates surge levels at Punta della Salute, while both emulators slightly underestimate, yet

350

375

still offer a closer match to observed amplitudes, indicating a consistent reduction in bias. In contrast, at Trieste, both emulators underestimate the peak significantly, while SHYFEM-MPI provides a better fit, underscoring the event- and site-specific variability in model performance under the given meteorological forcing.

For the second peak on December 28, all models, including SHYFEM-MPI, overestimate surge levels at Punta della Salute. Meanwhile, at Trieste, MLR-MADc² underestimates, and LSTMh-MADc² slightly overestimates the surge, highlighting the greater adaptability of the LSTM-based model to evolving surge patterns, albeit with occasional trade-offs in amplitude precision. Overall, these events show that MADc²-trained emulators can effectively track the evolution of complex surge events, though their accuracy at individual peaks remains sensitive to site characteristics, atmospheric conditions, and model architecture.

## 3.4 Application of the emulators to November 2022

To evaluate the generalization capability of the emulators beyond the training and testing period, we examined their performance during the November 2022 surge event, comparing them against both observed data (from the Punta della Salute station and from the CNR platform) and Med-Physics. This case offers a useful benchmark for assessing emulator behavior under unseen conditions, but it also highlights an important limitation: the MoSE flood barriers, operational since 2020, significantly altered surge propagation within the lagoon. Because neither the training data nor the Med-Physics simulations account for these barrier operations, discrepancies between emulator predictions and observations reflect changes in the physical system rather than shortcomings of the models themselves. In this sense, the November 2022 event illustrates both the potential and the boundaries of purely data-driven emulation, showing that while emulators can generalize beyond their training period, their predictive skill is constrained when external interventions fundamentally reshape hydrodynamics.

At Punta della Salute, both emulators, MLR-MADc² and LSTMh-MADc², closely track the timing and shape of the observed surge, outperforming Med-Physics in overall alignment (Figure 7a). The discrepancy in amplitude between model outputs and observed values is due to the activation of the MoSE flood barriers, which dampened the sea levels within the lagoon but are not represented in any of the emulators. Notably, because the emulators were trained on pre-MoSE data, their outputs rather resemble those recorded at the CNR platform, a nearby offshore location unaffected by MoSE, suggesting consistency in their learned representation of surge dynamics. In Trieste, model differentiation becomes clearer. LSTMh-MADc² provides a more accurate depiction of the magnitude of November 22nd peak, outperforming both the simpler MLR-MADc² and the Med-Physics simulation (Figure 8a). Whereas the MLR-based emulator tends to overestimate the peak.

The scatter plots (Figures 7b–d and 8b–d) confirm these results: both emulators outperform Med-Physics at the CNR platform and in Trieste, achieving superior metrics across the board. In this case LSTMh-MADc<sup>2</sup> (Figures 6d and 7d) consistently

emerges as the top performer among them, demonstrating stronger agreement with observed surges, despite minor underestimations in peak amplitude.

Figure 7: Emulator results for November 2022 at Punta della Salute. (a) Time series of observed and predicted storm surges; (b–d) Scatter plots comparing observed values (at the CNR platform) with reconstruction from: (b) Med-Physics model, (c) MLR-MADc<sup>2</sup> model trained on PS, and (d) LSTMh-MADc<sup>2</sup> model trained on PS.

Figure 8: Emulator results for November 2022 at Trieste. (a) Time series of observed and predicted storm surges; (b–d) Scatter plots comparing observed values with reconstruction from: (b) Med-Physics model, (c) MLR-MADc<sup>2</sup>, and (d) LSTMh-MADc<sup>2</sup>.

## 4 Discussion

385

395

An important consideration in this study concerns the use of PCA for reducing the spatial dimensionality of the predictors.

While PCA is effective in retaining the components that explain most of the variance, it applies a linear transformation to the input data, which may affect the emulator's ability to capture nonlinear patterns or interactions that are potentially relevant for storm surge dynamics.

Despite this potential limitation, the use of PCA still provides valuable insights into the relative importance of each predictor. As shown in Figures S9 and S10, the permutation importance analysis highlights a consistent hierarchy of predictors across both emulators and loss functions. For the full time series (Figure S9), sea surface height (SSH) emerges as the most influential feature, followed by y-component of wind stress (WSy) and the mean sea level pressure (MSLP). When focusing on surge peaks above the 99th percentile (Figure S10), the dominance of SSH as the leading predictor becomes even more pronounced, with WSy retaining secondary importance. In both the full time series and surge peaks, a few predictors exhibit slightly

© Author(s) 2025. CC BY 4.0 License.

EGUsphere Preprint repository

400 negative importance values. These should not be interpreted as physically detrimental influences but rather as artifacts of the permutation procedure, arising from resampling variability and finite sample effects when estimating small contributions.

An alternative to PCA is the use of encoding layers, such as those implemented in neural networks by Žust et al. (2021), Rus et al. (2023a), and Rus et al. (2023b), which can automatically learn nonlinear representations and extract higher-order dependencies from the predictors. In principle, such approaches could improve the ability of ML models to represent subtle spatial and temporal structures. In the present study, encoding layers were tested but yielded heavier models without improvements compared to PCA-based reduction. For this reason, PCA was retained as the preferred approach, offering a computationally efficient solution without a loss in predictive skill. Nonetheless, the potential of encoding layers remains promising, and future work could revisit these methods through optimized architectures or hybrid strategies that better balance model complexity and accuracy.

A key contribution of this work is the implementation of the MADc<sup>2</sup> loss function, a custom formulation that integrates the Mean Absolute Deviation (MAD) with a percentile-based term (MADp). This hybrid loss rewards both accurate timing of peaks and faithful reproduction of signal amplitude, making it better suited than MSE for learning and reproducing extreme events. The benefit of this formulation is further evidenced by the Probability Integral Transform (PIT) histograms (Figure 9), for which the MLR emulators were considered. PIT histograms provide a direct diagnostic of predictive calibration: a uniform distribution indicates that the emulator reproduces the full range of observed values without systematic under-or overestimation. While the MLR-MSE emulator shows deviations from uniformity (Figure 9a and 9c), especially under-representing the lowest and highest percentiles, the MLR-MADc<sup>2</sup> emulator produces a PIT histogram much closer to uniform (Figure 9b and 9d). This improvement demonstrates that the MADc<sup>2</sup>-based model not only enhances accuracy at extreme values but also yields better calibrated probabilistic predictions across the entire distribution.

430

Figure 9: Probability Integral Transform (PIT) histograms comparing the calibration of MLR emulators. Panels (a) and (c) show results with the MSE loss at Punta della Salute and Trieste, respectively, while panels (b) and (d) illustrate the improved uniformity achieved with the MADc<sup>2</sup> loss.

Across all tested architectures, emulators trained with MADc<sup>2</sup> consistently outperform their MSE-trained counterparts on metrics sensitive to surge extremes, such as SLF, MADp, and MADc (Figures 10 and 11). In contrast, emulators trained with the MSE loss achieve higher scores on conventional metrics like RMSE and Pearson correlation, yet systematically underestimate high-percentile surge values. This trade-off mirrors the results of Campos-Caba et al. (2024), who found that optimizing dynamic models for RMSE tends to favor simplified configurations (e.g., ERA5 + barotropic) that perform poorly on extremes, while penalizing more physically accurate setups.

Notably, the simple MLR emulator illustrates this trade-off well: when trained with MSE, it achieves excellent RMSE scores but performs worse than any other architecture on extremes; conversely, when trained with MADc², it attains some of the best performances for extreme events (Figure 10). Another relevant finding is the relatively poor performance of RNN-based models on extremes and their large skill variability when trained with MADc². RNNs are known to suffer from the vanishing gradient problem (e.g., Noh, 2021), which can hinder their convergence with a loss function like MADc², particularly when the minima of MAD and MADp do not coincide. Moreover, the inherent assumption in RNNs of a constant dependence of present conditions on past states may not align with storm surge dynamics, where dependence on past conditions weakens under extreme events. This may also explain the comparatively good extreme-event performance of simple feed-forward, stateless architectures like MLR, as well as gated architectures (e.g., LSTM and in particular LSTMh) that can quickly "forget" past states when conditions shift abruptly (Figure 11).

Figure 10: Scatter plots with performance metrics for SHYFEM-MPI and MLR models at Punta della Salute (a–c) and Trieste (d–f). Panels (b) and (e) show the MSE-based MLR models, while panels (c) and (f) correspond to the MADc²-based variants. Panels (a) and (d) show the performance of the SHYFEM-MPI model for reference.

460

Figure 11: Scatter plots with performance metrics for SHYFEM-MPI and LSTMh models at Punta della Salute (a–c) and Trieste (d–f). Panels (b) and (e) show the MSE-based MLR models, while panels (c) and (f) correspond to the MADc²-based variants. Panels (a) and (d) show the performance of the SHYFEM-MPI model for reference.

MADc² training shows clear improvements in both skill metrics and in reproducing the observed water-level distribution. At high percentiles, MADc²-trained emulators align more closely with the empirical distributions of tide gauge data. In Trieste, the best MSE-trained emulator underestimates the highest surge by more than 10 cm (Figure 12a), whereas MADc²-trained models reproduce the distribution more faithfully (Figure 12b). At Punta della Salute, the difference is subtler above the 99th percentile (Figures 12c–d); however, when considering values beyond the 99.9th percentile, it can be seen that the MADc²-based emulators improves with respect to the underestimations observed in the MSE-based models (Figures 12e–f). Nevertheless, for these higher surges (above the 99.9th percentile), the MADc² models produce some overestimations between the 99.9th and 99.98th percentiles.

Several MADc<sup>2</sup>-trained emulators match or even surpass the performance of SHYFEM-MPI, a high-resolution dynamical model specifically developed for storm surge prediction in the region, when the most extreme surges are considered. For instance, the distributions of bias, MADp, and MADc for MLR-MADc<sup>2</sup> for surge peaks above the 99<sup>th</sup> percentile shown in Figure 3 indicate that, over the 40 runs performed, this emulator outperforms SHYFEM-MPI across the mentioned metrics.

© Author(s) 2025. CC BY 4.0 License.

Similarly, most of the distributions for LSTMh-MADc<sup>2</sup> demonstrate better performance than the SHYFEM-MPI benchmark for the same evaluation criteria, including the selected runs based on the adopted validation-based model selection strategy. These results confirm that ML emulators, when trained with appropriate loss functions, can serve as efficient and accurate alternatives to physics-based models, particularly in ensemble forecasting or early warning applications.

Case studies of extreme events reinforce these results. The case studies of storm events from 1996, 2020, and 2022 provide valuable insight into the emulators' capacity to handle different surge dynamics and real-world complexities. The November 1996 event highlights the strength of MADc²-trained models (both MLR and LSTMh) in accurately capturing intense peaks, suggesting their effectiveness in learning the underlying physics from data — particularly when the events fall within the training distribution. In contrast, the December 2020 event underscores a trade-off: while the SHYFEM-MPI model better reconstructs the multi-peak structure at Trieste, the emulators still provide a more accurate estimate of the main peak, especially at Punta della Salute. This indicates that data-driven models may prioritize dominant features over finer-scale variability unless the latter is well-represented in training.

The November 2022 storm is especially revealing, as it occurred outside the training and testing periods and thus probes the models' generalization ability. Despite this, both MADc²-trained emulators outperform the physics-based Med-Physics model at the CNR platform and Trieste. This suggests a promising robustness in handling previously unseen conditions. However, the performance drop at Punta della Salute — likely due to MoSE flood barrier activations, which are not included among the input predictors — exposes a limitation: the lack of external forcings or human interventions in the training data can reduce local prediction accuracy. Importantly, emulators offer the flexibility to incorporate such additional predictors during training, which would enable the models to explicitly account for human interventions like barrier operations. Still, the fact that the emulators correctly track the timing and structure of the event, even under such altered conditions, points to a strong generalization capacity and resilience in preserving the underlying predictive relationships.

495

485

Figure 12: Cumulative Distribution Functions (CDFs) beyond the 99th percentile at Punta della Salute (a–b) and Trieste (c–d), for emulators trained with the MSE loss function (a, c) and the MADc² loss function (b, d). Panels (e) and (f) show the CDFs beyond the 99.9th percentile at Punta della Salute for MSE-based and MADc²-based models, respectively.

© Author(s) 2025. CC BY 4.0 License.

In terms of computational efficiency, the contrast between dynamical downscaling and the ML approach is striking. Dynamic downscaling required access to the high-performance computing facilities of the CMCC Supercomputing Center, where one year of simulation on the Zeus infrastructure—comprising 348 Lenovo SD530 dual-processor nodes (12,528 cores in total) interconnected via an InfiniBand EDR network, with a theoretical peak performance of 1.202 TFlops—took approximately 36 hours to complete when executed on 36 cores. By comparison, the ML emulators could be trained and validated on a single high-end laptop equipped with an Intel® Core™ Ultra 9 processor, 64 GB of RAM, and an NVIDIA RTX™ 3000 Ada GPU. On this system, the total runtime for a single experiment was reduced from hours to seconds or minutes, with average execution times ranging from 20–40 s for MLR and 20–60 s for MLP models to under two minutes for RNN-based architectures and 190–400 s for the most computationally demanding LSTMs. This difference highlights the ability of ML-based emulators to achieve orders-of-magnitude improvements in efficiency without the need for supercomputing infrastructure.

## 510 **5 Conclusions**

515

520

This study demonstrates the potential of machine learning emulators as flexible and efficient tools for storm surge prediction in coastal regions. A central contribution is the development of the MADc² loss function, specifically designed to improve the emulation of extremes by jointly rewarding amplitude accuracy and percentile alignment. Emulators trained with MADc² consistently outperformed both their MSE-trained counterparts and a state-of-the-art numerical model (SHYFEM-MPI), representative of the best currently achievable with physics-based approaches, in reproducing extreme peaks, while maintaining acceptable accuracy for the full time series.

Our experiments highlight that model architecture is less critical than the choice of loss function: even simple approaches such as MLR, when trained with MADc<sup>2</sup>, delivered skill on extremes comparable to and even exceeding more complex neural networks. LSTMh architectures also performed strongly, whereas RNN-based models showed comparatively weaker results, likely linked to vanishing gradient issues and their assumption of constant temporal dependence. The simplicity and computational efficiency of MLR MADc<sup>2</sup> make it particularly attractive for large-scale or global studies, where rapid training and interpretability are key advantages.

- MADc<sup>2</sup>-trained emulators also proved robust in generalization, retaining strong predictive skill during the November 2022 event despite operating outside their training and validation periods. This result indicates that, when trained with objectives tailored to extremes, ML emulators can extend their skill to previously unseen conditions, a crucial requirement for early warning and climate resilience applications.
- Overall, this work reinforces the promise of data-driven emulators as complementary or standalone components of operational coastal hazard forecasting systems. The fact that they rival a top-tier numerical model underscores their readiness for practical

© Author(s) 2025. CC BY 4.0 License.

535

545

555

EGUsphere Preprint repository

integration. While the fitted cases here rely on observational data, whose availability can be a limiting factor, the emulators could also be trained directly on numerical model outputs, thereby extending their applicability to locations without long instrumental records. Such flexibility would enable seamless implementation within operational forecasting chains, including the possibility of probabilistic ensemble predictions. With further refinement and possibility to easily include additional forcings where needed (e.g., the MoSE barriers in Venice), these approaches offer a scalable pathway toward next-generation multihazard prediction frameworks that are adaptive, computationally efficient, and better aligned with the needs of risk management and preparedness.

6 Data availability

540 The raw data supporting the conclusions of this article will be made available by the authors without undue reservation.

7 Author contribution

RCC implemented the ML emulators, conducted the training, validation, and testing processes, performed post-processing and performance evaluation, and prepared the manuscript. LM supervised the implementation of the ML emulators, post-processing, and performance evaluation, and contributed to manuscript preparation. PC contributed theoretical insights during the implementation of the ML emulators, performance evaluation, and manuscript preparation. AM contributed to the performance analysis and manuscript preparation. MV, MT, IF, and SC contributed to the preparation and revision of the manuscript.

**8** Competing interests

Co-author Massimo Tondello is employed by the company HS Marine SrL. Co-author Michalis Vousdoukas is employed by the company MV Coastal and Climate Research Ltd. The remaining authors declare that the research was conducted in the absence of any commercial or financial relationship that could be construed as a potential conflict of interest.

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
