# Peer review of "Storm surge dynamics in the northern Adriatic Sea: comparing AI emulators with high-resolution numerical simulations"

_EGUsphere, 2025_

## Referee Comment (RC1)

Summary
The authors trained and evaluated the performance of a suite of data-driven models at predicting extreme storm surges at two locations in the Adriatic Sea. This is a challenging and relevant issue, and I therefore think that this study is, in principle, useful and timely. However, considering recent studies on the same issue, I think the authors could take a few additional steps to really contribute to helping the science forward (please see below). Additionally, some methodological aspects need to be clarified to better support the authors' current conclusions.

Main comments

1. According to the authors, tailoring their loss function to the extremes is a key contribution of the manuscript (L411). Recent studies have also shown the benefits of adapting the loss function to predict extremes, using either density-based weights (Hermans et al., 2025) or quantile loss (Longo et al., 2025). As the MADc$^2$ loss function is a similar intervention, the better performance obtained is not super surprising. It would therefore be very helpful if the authors could also test other loss functions that address the data imbalance, so that in addition to confirming previous findings, they could also draw conclusions about which loss functions are most effective in their case.

2. The authors find that their multi-linear regression model performs similarly to some of their neural networks. However, due to several methodological aspects is not fully clear whether the neural networks were appropriately optimized to support this conclusion:

   a. The authors applied PCA to reduce the dimensions of the input data and did not consider convolutional layers in their architectures. Tiggeloven et al. (2021) and Hermans et al. (2025), however, found that for larger prediction regions, convolutional layers are beneficial for the prediction of (extreme) storm surges. Adding a clear caveat to the conclusions in L519-524 to reflect on this would be helpful.
   b. The authors did not specify the number of preceding timesteps at which predictor data was used for the LSTMs. If no look-back window was used, this would likely lead to a worse (relative) performance.
   c. The authors optimized several of their input settings for all model architectures based on tests with only the MLR model (L104-108). Could the authors motivate whether these settings are also expected to work best for their other architectures, and if not, reflect on this in the discussion?
   d. Hyperparameter tuning is mentioned only briefly. Could the authors add additional detail on what parameters they used (not just the depth of the network, but also the other learning parameters) and to what extent these parameters were optimized? These are important details to place their results into context.

3. The authors used sea surface height simulations from a high-resolution ocean reanalysis (Med-MFC) as a predictor variable for their data-driven models. I do

not understand the rationale for this: if used in a forecasting or longer-term prediction setting, this means that to obtain the performance reported by the authors, a high-resolution hydrodynamic model will be necessary as well, which defeats the purpose of the data-driven models. Furthermore, unless I misunderstand, they use part of what they want to predict (surge) as input (sea surface height), meaning that performance of the data-driven models will be enhanced by construction. Could the authors please explain their methods in this regard and discuss how this affects their conclusions in L465-472? How would the data-driven models perform using only atmospheric data as predictors, as previous studies did?

4. L540: The authors did not make their code and data available, so their results are currently irreproducible and cannot be reviewed. This needs to be published in an appropriate repository.

Other comments

L35: Calafat et al., 2022 did not attribute the trends they found to climate change, so I suggest using a different reference here.

L40: After 'widely' I would expect a few more examples, such as Tadesse et al. (2020) and references therein.

L48-49: Yes, but they also showed that convolutional layers did help to increase performance when using larger predictor regions. Please also see Hermans et al. (2025).

L61: As only two locations are considered, point (2) is not really addressed in this manuscript either.

L104: Please consider showing the spatial domain of predictors in the maps in Figure 1 as well.

Section 2.4: Details about the architecture, number of layers, units, other hyperparameters (such as dropout rate, learning rate, etc.) and their optimization are absent -> please provide for reproducibility.

L163: The test split only covers 3 years -> are the extremes during this period representative of the overall distribution? Could they be easier or more difficult to predict than extremes in other 3-year periods by chance? Did the authors test different splits or split ratios?

L173-174: Please explain how these settings were determined.

L181: Why specifically was the 99$^{th}$ percentile used? The results of Harter et al. (2024) and Hermans et al. (2025) suggest that for even more extreme values, the performance of data-driven models falls off compared to a high-resolution hydrodynamic model. It would be interesting to check whether this is also the case here.

Section 4 (Discussion): comparison with and discussion in the context of previous literature on data-driven modeling of storm surges (the ones mentioned above and several other relevant studies) is largely missing; I would encourage the authors to describe how their results fit into the bigger picture and help advance the field.

General: please consider using perceptually uniform instead of rainbow colormaps.

Additional references:
Hermans et al. (2025): https://nhess.copernicus.org/articles/25/4593/2025/
Longo et al. (preprint): https://essopenarchive.org/users/952724/articles/1322927-a-deep-learning-framework-for-extreme-storm-surge-modelling-under-future-climate-scenarios
Tadesse et al. (2020): https://www.frontiersin.org/journals/marine-science/articles/10.3389/fmars.2020.00260/full